# CURRICULUM-GUIDED LAYER SCALING FOR LANGUAGE MODEL PRETRAINING

## ABSTRACT

As the cost of pretraining large language models grows, there is continued interest in strategies to improve learning efficiency during this core training stage. Motivated by cognitive development, where humans gradually build knowledge as their brains mature, we propose Curriculum-Guided Layer Scaling (CGLS), a framework for compute-efficient pretraining that synchronizes increasing data difficulty with model growth through progressive layer stacking (i.e., gradually adding layers during training). At the 100M parameter scale, using a curriculum transitioning from synthetic short stories to general web data, CGLS outperforms baseline methods on the question-answering benchmarks PIQA and ARC. Pretraining at the 1.2B scale, we stratify the DataComp-LM corpus with a DistilBERT-based classifier and progress from general text to highly technical or specialized content. Our results show that progressively increasing model depth alongside sample difficulty leads to better generalization and zero-shot performance on various downstream benchmarks. Altogether, our findings demonstrate that CGLS unlocks the potential of progressive stacking, offering a simple yet effective strategy for improving generalization on knowledge-intensive and reasoning tasks.

## 1 INTRODUCTION

Large language models (LLMs) are typically pretrained in a single, continuous pass, processing all tokens with a uniform amount of computation regardless of their complexity or relevance to downstream tasks of interest. While this approach has shown remarkable success in large-scale models like GPT-4 (OpenAI et al., 2023) and Llama 3 (Dubey et al., 2024), it differs significantly from how humans learn, often leading to models that excel in generating coherent text but struggle with long-context reasoning across varied tasks (Schnabel et al., 2025). Recent works like Phi-3 (Abdin et al., 2024), MiniCPM (Hu et al., 2024), and others (Feng et al., 2024) have explored *midtraining*, adjusting the training data distribution partway through training by incorporating higher-quality, multilingual, or long-form text. However, this coarse-grained curriculum is applied on *fixed* model architectures. Inspired by how humans progressively build knowledge alongside their physically growing brains, we explore whether gradually scaling a model in tandem with increasingly complex data can enable more efficient and effective learning.

Curriculum learning (Bengio et al., 2009) has shown success in guiding models from easier to harder tasks, but remains largely unexplored in modern language model (LM) pretraining. Recent studies propose several alternative methods to schedule complexity and model scale during LM training. Fan & Jaggi (2023) train small proxy models to determine sample learnability, enabling efficient complexity-aware curricula for training LMs. Progressive layer stacking (Gong et al., 2019; J. Reddi et al., 2023; Saunshi et al., 2024; Gu et al., 2020; Yang et al., 2020; Panigrahi et al., 2024), layer-dropping (Zhang & He, 2020), and pruning (Kim et al., 2024) are gradual architectural adjustments that accelerate training and reduce inference cost. In computer vision, progressive model growing has been explored for image generation (Karras et al., 2017) and segmentation (Fischer et al., 2024).

In particular, progressive stacking is an approach that trains a model in stages, adding new layers (randomly initialized or copied) at the start of each stage until a final model size is eventually reached. Despite the intuitive appeal of progressive stacking as an approach to learn hierarchical features and improve compute efficiency, recent studies suggest it underperforms in knowledge-intensive tasks such as closed-book question answering. For example, MIDAS (Saunshi et al.,

2024) finds that progressively grown models lag behind full-capacity models trained from scratch on memorization-focused tasks. In our own setup and compute-controlled experiments, we also find that progressive growing performs similarly or worse than standard training on several benchmarks.

We hypothesize that model expansion alone is insufficient for compute-efficient learning of hierarchical features, and investigate whether stacking should be paired with a structured learning signal to realize its potential. Inspired by cognitive development, we propose **Curriculum-Guided Layer Scaling (CGLS)**, a pretraining paradigm that couples progressive model expansion with a data curriculum of gradually increasing complexity. To preserve and build on previously learned representations during model growth, CGLS uses staged training: at each step, newly added transformer layers are first trained in isolation (with earlier layers frozen), before the entire model is fine-tuned on a more complex data distribution. This strategy protects previously learned representations during capacity increases and aligns model complexity with data difficulty throughout training.

**Our Contributions.** In summary, we make the following contributions:

- We propose **Curriculum-Guided Layer Scaling (CGLS)**, a unified framework for scaling both model capacity and data complexity during pretraining, combining layerwise expansion and stage-wise transfer with curriculum learning.
- We conduct a rigorous empirical evaluation of CGLS at two parameter counts and compute scales, demonstrating its scalability.
  - At a GPT-2-Small parameter count using a synthetic-to-webtext curriculum, CGLS achieves consistent gains in perplexity and downstream QA performance (1.0% on average, and +2.1% on PIQA).
  - At a LLaMA-3.2-1B parameter count and 2.5B tokens, we stratify the DataComp-LM (DCLM) corpus (Li et al., 2024) using a DistilBERT-based classifier (Sanh et al., 2019) and show that CGLS outperforms compute-matched baselines on knowledge-intensive and reasoning-heavy tasks, on average by 1.70% across all benchmarks and by as much as 5% on ARC-Easy. Scaling up to a Chinchilla-optimal 20B tokens (Hoffmann et al., 2022), gains are even larger, with an average 3.90% increase across benchmarks.

Overall, we demonstrate that joint scaling of model and data complexity unlocks the benefits of progressive stacking for language model pretraining, yielding better downstream performance on knowledge-intensive tasks up to the 1B parameter and 20B tokens scale.

## 2 RELATED WORK

Curriculum learning, formalized by Bengio et al. (2009), has shown empirical benefits in many domains, from vision (Hacohen & Weinshall, 2019; Weinshall et al., 2018) to reinforcement learning (Graves et al., 2017). However, it remains underexplored in the context of large language model (LLM) pretraining, where most models are trained in a single pass over a diverse and unstructured dataset. Self-paced learning strategies (Kumar et al., 2010) adjust training dynamics based on model uncertainty, gradually introducing harder samples as the model becomes more confident. More recently, efforts such as DataComp (Li et al., 2024), MetaCLIP (Xu et al., 2023), and Tower (Alves et al., 2024) demonstrate that large performance gains can arise from careful dataset curation, even without architectural changes. Related work like Thrush et al. (2024) and Ankner et al. (2024) investigates the use of small language models to identify high-quality data subsets that better align with LLM performance, but does not explore staged training. Likewise, Gururangan et al. (2020) show the value of domain-specific continued pretraining after generic training, indicating that structured data exposure plays a key role even in later stages. A closely related curriculum-learning baseline is Beyond Random Sampling (BRS) (Zhang et al., 2025), which evaluates curricula with ten difficulty groups and finds that fine-grained groupings provide little advantage over a simpler 3-tier curriculum. Still, these approaches do not consider whether data curricula coordinated with model growing could improve compute efficiency.

Parallel to this, a growing body of research examines how to progressively increase model capacity during training. Soviany et al. (2021) and others have noted that the curriculum principle may apply not to data, but also to model size. In vision, progressive training has been used to stabilize training and reduce compute costs, e.g., by gradually growing generative networks (Karras et al., 2017) or

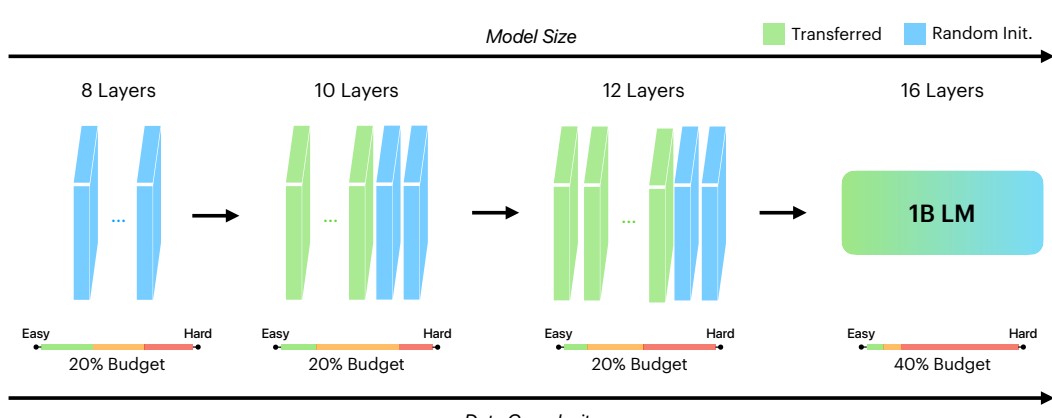

Figure 1: **Curriculum-Guided Layer Scaling (CGLS)** is a new paradigm for compute-efficient language model pretraining that grows data complexity and model depth in tandem. We illustrate CGLS for a Llama-3.2-1B scale model with four training stages. Training begins with an 8-layer model on a data split consisting equally of data from all levels (high-school, undergraduate, and graduate). The learned weights from this stage are transferred to a larger 10-layer model, freezing the pretrained weights and training the new layers on a small, balanced data split for better initialization. The entire model is then unfrozen and pretrained on the more difficult data split. This process is repeated until the target model scale is reached.

using function-preserving transformations (Chen et al., 2015). For Transformers, techniques such as LayerDrop (Fan et al., 2019) and Width-Adaptive Transformers (Zhao et al., 2024) enable dynamic capacity at inference time, though they are not designed for staged training. Mixture-of-Experts (MoE) models (Shazeer et al., 2017; Fedus et al., 2022) similarly allocate compute conditionally across tokens but often suffer from expert collapse or shallow specialization to token-level features (Zoph et al., 2022). Alongside the vast progressive stacking literature mentioned in Section 1, more recently, architectural methods have emerged to progressively grow models during training. Wang et al. (2023) propose learning an explicit mapping from a smaller transformer's parameters into a larger one, enabling weight reuse and transfer. Complementary to this, Yao et al. (2023) introduce smoother transitions between training stages by gradually increasing the influence of new parameters. Du et al. (2024) compare the various stacking strategies in prior work, and demonstrate their scalability and optimal usage. Such efforts underscore the promise of synchronized curricula for data and model: gradually increasing data complexity or domain diversity in tandem with growing model capacity or specialization. Our Curriculum-Guided Layer Scaling approach unifies prior ideas of curriculum learning and progressive model scaling into a single coordinated pretraining strategy.

## 3 METHOD

Humans do not learn language by memorizing the full dictionary on day one. Instead, we begin with simple patterns: repetitive sounds, basic grammar, and simple visual inputs, and gradually expand to handle more abstract concepts, longer sentences, and complex reasoning as our cognitive capacities mature. This progression in data complexity is tightly coupled with the brain's developmental growth: as new neural structures form, they scaffold increasingly sophisticated representations of the world (Kolk & Rakic, 2022). We study whether this notion of growing neural and data complexity in tandem can improve the compute efficiency of language model pretraining.

Previous work has explored increasing either neural or data complexity alone during language model pretraining, through curriculum learning and various approaches for expanding neural nets; we review these approaches in Section 2 above. Inspired by the structured development of the human brain, we develop and study a pretraining paradigm that synchronizes a data curriculum with the

gradual addition of randomly initialized layers to the language model. We next describe these two components in our framework—curriculum and layer expansion—in turn.

**Curriculum Learning.** To construct a pretraining curriculum, we organize documents into progressively harder splits, beginning with simpler and more structured samples before introducing more complex, diverse, and noisier data. Let $\mathcal{D}_i$ represent the dataset used at stage $i$, where $|\mathcal{D}_i|$ is the number of tokens. We first segment our pretraining dataset into easy, medium, and hard components $\mathcal{D}_{\text{Easy}}, \mathcal{D}_{\text{Medium}}$ and $\mathcal{D}_{\text{Hard}}$. Then, the data distribution for stage $i$ can be expressed as a mixture of the easy, medium, and hard components with mixture weights $(p_i, q_i, r_i), p_i + q_i + r_i = 1$:

$$\mathcal{D}_i = p_i \cdot \mathcal{D}_{\text{Easy}} + q_i \cdot \mathcal{D}_{\text{Medium}} + r_i \cdot \mathcal{D}_{\text{Hard}}.$$

The mixture weights $(p_i, q_i, r_i)$ are adjusted to balance each stage's difficulty, as shown in Figure 1.

**Layer Expansion.** In parallel with a structured curriculum, we expand the model's capacity at each stage by incrementally adding transformer layers. Beginning with a shallow model (e.g., 6 layers), we train on simpler datasets before transferring the learned weights to a larger model with additional layers. Let $\boldsymbol{\Theta}^{(i)}$ denote the parameters of the model at stage $i$, and $\boldsymbol{\Theta}^{(i)}_{1:k}$ the parameters of the first $k$ non-embedding layers. Then, at stage $i + 1$, the parameters are initialized as:

$$\boldsymbol{\Theta}^{(i+1)} := \{\boldsymbol{\Theta}^{(i)}_{\text{embed}} \circ \boldsymbol{\Theta}^{(i)}_{1:k} \circ \boldsymbol{\Theta}^{\text{random}}_{k+1:n} \circ \boldsymbol{\Theta}^{(i)}_{\text{norm}} \circ \boldsymbol{\Theta}^{(i)}_{\text{lm}}\}$$

where $\boldsymbol{\Theta}^{\text{random}}_{k+1:n}$ represents the newly added, randomly initialized layers. Following the results of Kumar et al. (2022), training is then performed in two phases:

1. **Initialization phase:** the new parameters $\boldsymbol{\Theta}^{\text{random}}_{k+1:n}$ are trained, while all other layers are frozen; that is, the embedding layer $\boldsymbol{\Theta}^{(i)}_{\text{embed}}$, previous non-embedding layers $\boldsymbol{\Theta}^{(i)}_{1:k}$, normalization layer $\boldsymbol{\Theta}^{(i)}_{\text{norm}}$, and language modeling head $\boldsymbol{\Theta}^{(i)}_{\text{lm}}$ are all kept frozen.

2. **Full tuning phase:** all parameters in $\boldsymbol{\Theta}^{(i+1)}$ are optimized.

This approach aims to ensure that the representations learned in prior layers are effectively utilized and not overwritten when the model size is increased. This progression is illustrated in Figure 1.

## 4 EXPERIMENTS

The core intuition of our approach for compute-efficient pretraining is to follow a slow transition from highly structured to more diverse data while scaling model capacity. We construct pretraining curricula using the TinyStories (Eldan & Li, 2023), BookCorpus (Zhu et al., 2015), and DataComp for Language Models (DCLM) (Li et al., 2024) corpora. We first implement our approach at a GPT-2 Small scale (Radford et al., 2019), starting with a small model trained on simple, synthetic language patterns (TinyStories) and gradually increasing both data complexity and model size. Then, we scale our approach to the parameter count of Llama-3.2-1B (Dubey et al., 2024), applying it to a stratified version of the DCLM dataset, where document complexity levels (e.g., high school, undergraduate, graduate) are inferred via a DistilBERT classifier trained on a small set of GPT-4o-labeled samples.

Next, we describe this particular instantiation of data curricula that we used in our experiments.

### 4.1 DATA CURRICULA

In conventional LLM pretraining, noisy and diverse datasets (e.g., Common Crawl or The Pile) are used early, and high-quality curated datasets are reserved for the final stages (Hu et al., 2024), or fine-tuning (Ouyang et al., 2022). This approach assumes that early exposure to broad linguistic patterns enhances generalization, while later fine-tuning reduces noise artifacts (Brown et al., 2020). In contrast, we explore whether incorporating more structure in early stages and later transitioning to diverse corpora can improve downstream performance. We find that different curricula work best at our two different scales and describe them in depth next.

**GPT-2-Small Scale Curriculum.** For the GPT-2 experiments, we start with a split containing samples primarily from TinyStories, consisting of synthetically-generated short stories comprehensible to a young child. The goal of training on such highly-structured data is to establish robust low-level linguistic representations. We then gradually introduce more challenging datasets, such as BookCorpus and DCLM, characterized by broader vocabulary, complex sentence structures, and diverse contexts. Our hypothesis is that early exposure to clean, consistent patterns helps the model develop reliable embeddings, which later support generalization to noisier data such as DCLM.

**Llama-3.2-1B Scale Curriculum.** At the 1B scale, rather than using separate datasets of increasing complexity, we stratified the DCLM-Baseline-1.0 corpus itself into tiers of general, domain-specific, and specialized reasoning complexity. This enabled us to perform curriculum progression entirely within a single large corpus, while leveraging the model's increased capacity to handle more complexity earlier in training as compared to the small-scale experiments. To implement this, we first perform *complexity stratification* using a separate pretrained LLM:

1. **Constructing Classifier Training Set:** Using GPT-4o-mini, we label a subset of 20,000 documents by classifying complexity into three levels:
   - *High School*: Structured text with limited abstraction or formal reasoning, such as news articles, beginner-level Wikipedia pages, or simple conversational stories.
   - *Undergraduate*: Content with moderate reading difficulty and some domain-specific concepts, including ArXiv abstracts and science articles that require basic abstract reasoning.
   - *Graduate/Advanced*: Highly technical or specialized content such as academic papers in AI/ML, legal documents, medical research studies, and code repositories (e.g., GitHub).

2. **Class Balancing & Splitting:** We then upsampled any underrepresented categories, and performed a random 80/10/10 split for training, validation, and testing, respectively.

3. **Model-Based Classification:** Finally, we trained a simple DistilBERT-based classifier (Sanh et al., 2019) on this data, which achieved $> 90\%$ test-set accuracy and was used to label the full 2M document subset of DCLM used for training. Then, by choosing mixture weights over the stratified splits of DCLM, our data schedule (see Appendix A.1) reflects a gradual shift from general to specialized content as model depth increases.

### 4.2 Compute-Matched Method Comparisons

In our experiments at the GPT-2-Small and LLaMA-3.2-1B parameter counts, we fix a compute budget in pretraining FLOPs across all methods to enable clean comparisons. Prior work on LLM scaling suggests that optimal training lies near the "Chinchilla" compute/data tradeoff curve (Hoffmann et al., 2022), i.e., with $20\times$ more tokens than parameters. In our default experiments, we select a lower FLOP budget due to academic compute constraints, using 700M tokens for the GPT-2-Small scale experiments and 2.5B tokens for our initial Llama-3.2-1B scale experiments. Then, we perform a focused larger-scale investigation of our method at the Chinchilla-optimal ratio of 1B parameters and 20B tokens, in addition to a 4B-token domain-shift experiment transitioning from general text to code. Following well-established practices in the LLM scaling laws literature (Kaplan et al., 2020), we approximate the total compute budget for each experiment, in FLOPs, as:

$$\text{Total FLOPs} = 6 \cdot \text{Tokens} \cdot \text{Parameters} \cdot \text{Epochs}$$

We next detail the compute-matched configurations of CGLS that we evaluate, before discussing a set of natural baseline approaches applying curricula over data and model complexity.

**CGLS Compute-Matched Configurations.** Our experiments consisted of training several compute-matched models under our CGLS framework, with varying dataset splits and starting number of layers $N_1$. These are summarized as follows, with detailed data splits in Appendix A.1.

- **Simple (GPT-2-Small scale only):** Starting from a randomly-initialized 6 layer model, we train for one epoch on 111M tokens from TinyStories. These weights are transferred to a 9 layer model, which is trained twice on 140M tokens of BookCorpus: once in the initialization phase (transferred layers frozen) and once in the full tuning phase (all layers unfrozen). We apply a similar progression at 12 layers, using 200M DCLM tokens for the initialization phase and 461M tokens for the full tuning phase.

- **CGLS:** Training begins with $N/2$ layers, where $N$ is the desired final model depth, trained on a curriculum-balanced dataset, followed by a staged progression to the full model size. At each stage, newly added layers are first trained with a fixed data split (used for all initialization phases), before the full model is fine-tuned on a progressively more challenging data distribution.

**Compute-Matched Baselines.** For both model scales, we evaluate the following compute-matched baselines under the same pretraining FLOP budget:

- **Randomized baseline:** A model with the final depth (a "full model") is trained from scratch on a randomized mixture of the available data.
- **Curricularized baseline:** This baseline simulates a coarse curriculum by training the full model sequentially on increasingly complex data subsets. For GPT-2-Small, we train on TinyStories, then BookCorpus, then DCLM. For LLaMA-3.2-1B, we follow the same staged training but use the stratified DCLM splits (high school, undergraduate, graduate). Both this curriculurized baseline and CGLS use the same difficulty classifier, the same three difficulty tiers, and the same pacing schedule; all optimization hyperparameters and compute budgets are matched. The only difference is whether the model depth is expanded during training.
- **Layer-scaling only:** This baseline incrementally expands the model depth across stages, in the same schedule and manner as CGLS, but trains on the full (unstructured) data distribution at each stage without a curriculum. This isolates the contribution of progressive stacking alone.
- **MIDAS:** MIDAS progressively stacks layers across training stages but does not control the complexity of the training data. We reimplemented the method from pseudocode provided in the original paper (Saunshi et al., 2024), as there is no public codebase, with PROP-3 stage budgets, a block size of 3 for GPT-2-Small, and a block size of 4 for LLaMA-3.2-1B. Layer weights from previous stages were copied at each model expansion, as in the original paper, and FLOPs were matched with the other baselines and CGLS.

**Further CGLS Implementation Details.** At GPT-2-Small parameter scale, we scaled from 6 to 12 layers in 3 stages; at Llama-3.2-1B parameter scale, from 8 to 16 layers in 4 stages. We found that this starting depth yielded optimal results, as explored in the Appendix A.3. The train FLOP budgets for each stage were also optimized and allocated to favor later, larger stages (20% each for the first three stages; 40% for the final stage at Llama-3.2-1B scale), ensuring sufficient resources for fine-tuning on the most complex data. Additionally, we allocated 20% of the FLOPs for each stage to training only the new layers, and 80% for fine-tuning the full-unfrozen model. Further details on our experimental setup and ablation studies of these hyperparameters are provided in the Appendix A.1, A.3.

## 4.3 Evaluation

We evaluate the generalization and reasoning capabilities of the models on several downstream tasks:

- **PIQA (Physical Interaction QA)** (Bisk et al., 2019): This dataset tests physical commonsense reasoning by presenting multiple-choice questions about everyday scenarios.
- **ARC Easy and Challenge** (Clark et al., 2018): The AI2 Reasoning Challenge benchmarks measure a model's ability to answer elementary and middle-school science questions. The "Easy" subset contains relatively straightforward questions, while the "Challenge" subset consists of harder, multi-step reasoning tasks.
- **HellaSwag** (Zellers et al., 2019): This task evaluates common-sense reasoning and next-word prediction, requiring the model to identify the most plausible continuation for a given context. Given our constrained academic compute budget, raw accuracy on multiple-choice tasks tends to have low signal-to-noise (Heineman et al., 2025). Instead, we evaluate *Token-Normalized Accuracy* (Kydlíček et al., 2024; Gu et al., 2024), a metric that better accounts for length and bias effects in generative modeling to obtain a low-variance signal at small scales. This is defined as:

$$\text{acc}_{\text{token}} = \arg\max_i \frac{\ln P(a_i \mid q)}{\texttt{num\_tokens}(a_i)}$$

- **LAMBADA** (Paperno et al., 2016): An open-book language modeling task designed to assess a model's ability to perform long-range dependency reasoning. We report zero-shot accuracy, computed as whether the model correctly predicts the final word of a passage.

At our largest experiment scale of 1B parameters trained on 20B tokens, we additionally test the following benchmarks:

- **GPQA (Graduate-Level Physics QA)** (Rein et al., 2023): A challenging benchmark designed to assess high-level scientific and mathematical reasoning. Questions are drawn from graduate physics coursework and require multi-step conceptual understanding.
- **TLDR9+ (Summarization)** (Sotudeh et al., 2021): A summarization task based on Reddit posts, requiring abstractive generation of concise summaries from long, noisy inputs. We report Rouge-L (Lin, 2004) following prior work.
- **OpenRewriteEval (ORE)** (Shu et al., 2023): An open-ended rewriting and paraphrasing benchmark. The model must rewrite user-provided text to satisfy semantic constraints such as preserving meaning, altering tone, or improving clarity. We report Rouge-L (Lin, 2004) here as well.
- **InfiniteBench EN.QA (Long-Context QA)** (Zhang et al., 2024): A long-context benchmark where models must answer questions requiring retrieval and reasoning over very lengthy contexts. We report the English QA subset (EN.QA), which stresses information retention and retrieval at sequence lengths far beyond typical context windows. We report ROUGE F1 scores as specified for this benchmark.

All tasks are evaluated in a zero-shot setting to measure base model quality without additional fine-tuning, following recent academic pretraining works such as Hwang et al. (2025), although we include some few-shot results in the Appendix A.2. We report the token-normalized accuracy, as described above, for all multiple-choice tasks. We additionally compute perplexity on a 180,000 sample subset of the Pile dataset, capturing how well the methods model diverse, noisy data from a variety of domains. We excluded LAMBADA at the GPT-2-Small scale, as our models at this compute budget perform at or near chance on this benchmark, making it uninformative for comparing pretraining strategies.

## 5 RESULTS

Our experiments evaluate the performance of the CGLS training strategy at two parameter counts (matched to GPT-2-Small and Llama-3.2-1B) across perplexity and zero-shot accuracy on several downstream tasks. Table 1 summarizes the findings, comparing CGLS to several baselines with matched FLOP budgets. Across both model scales, GPT-2-Small trained on 700M tokens and Llama-3.2-1B trained on 2.5B tokens, CGLS consistently outperforms baseline approaches on reasoning-focused benchmarks such as ARC and PIQA, demonstrating its effectiveness in improving generalization under fixed compute budgets. CGLS shows particular strength in structured and knowledge-intensive tasks, supporting the benefits of aligning model growth with data complexity.

**GPT-2-Small Scale.** The GPT-2 scale CGLS model achieves consistent improvements on downstream tasks, and consistently lower perplexity across nearly all subsets of The Pile, as illustrated in Table 1 and Section A.6. For example, CGLS attains a 60.12% accuracy on PIQA, a 2.1-point improvement over the randomized baseline, along with gains on ARC-Challenge (22.53% vs. 21.08%).

**LLaMA-3.2-1B Scale.** The larger models demonstrate clearer trends. While perplexity results diverged from the GPT-2-Small trend, with the randomized baseline achieving the lowest perplexity across all Pile domains (see A.6 for detailed perplexity results), CGLS nonetheless achieves the strongest downstream performance. Across nearly all downstream tasks, CGLS improves over the best baseline: e.g., on PIQA (61.21% vs. 59.09%), ARC-Easy (41.37% vs. 36.36%). The gains on ARC-Easy are especially striking, exceeding the best baseline by over 5 percent. **Notably, the stacking baselines, without a synchronized data curricula, did not improve downstream performance.**

| Model | The Pile↓ | PIQA↑ | ARC-E↑ | ARC-C↑ | HellaSwag↑ | Lambada↑ | Average↑ |
|---|---|---|---|---|---|---|---|
| **GPT-2-Small** | | | | | | | |
| **Pretrained GPT-2-Small**⋆ | 23.97 | 62.51% | 39.48% | 22.70% | 31.14% | — | 38.96% |
| Baseline (Randomized) | 37.51 | 58.00% | 36.11% | 21.08% | **26.60**% | — | 35.45% |
| Baseline (Curricularized) | 36.09 | 59.09% | 36.78% | 22.18% | 26.43% | — | 36.12% |
| Layer Scaling Only | 39.67 | 58.76% | 35.82% | 21.42% | 26.35% | — | 35.59% |
| MIDAS (Saunshi et al., 2024) | 49.90 | 56.91% | 34.81% | 21.16% | 25.66% | — | 34.64% |
| CGLS (Simple) | 35.93 | 59.03% | **37.67**% | 21.76% | 19.50% | — | 34.49% |
| CGLS | **34.33** | **60.12**% | 36.62% | **22.53**% | 26.55% | — | **36.46**% |
| **Llama-3.2-1B** | | | | | | | |
| **Pretrained Llama-3.2-1B**⋆ | 9.25 | 74.32% | 60.56% | 36.52% | 63.67% | 62.00% | 59.41% |
| Baseline (Randomized) | **20.83** | 59.09% | 36.36% | 24.24% | 34.20% | 32.99% | 37.38% |
| Baseline (Curricularized) | 21.93 | 58.27% | 35.19% | **25.43**% | **34.65**% | **33.09**% | 37.33% |
| Layer Scaling Only | 29.53 | 57.45% | 32.58% | 23.46% | 30.00% | 27.73% | 34.24% |
| MIDAS (Saunshi et al., 2024) | 27.64 | 55.44% | 32.58% | 24.23% | 29.62% | 25.87% | 33.55% |
| CGLS | 25.34 | **61.21**% | **41.37**% | **25.43**% | **34.65**% | 32.74% | **39.08**% |

Table 1: Evaluation results comparing CGLS and baseline approaches across average perplexity on The Pile dataset and core reasoning- and knowledge-intensive downstream tasks. CGLS improves accuracy on downstream tasks like PIQA compared to the baselines. All models are compute-matched, but utilize much less compute than the pretrained variants of these models. ⋆Not compute-matched to the other methods, and trained on orders-of-magnitude more data. We provide pretrained results to illustrate a rough oracle upper bound for the given architecture. The best method among the compute-matched approaches is bolded, with the second-best method underlined.

## 5.1 EXTENDING CGLS

**Scaling to a 1B Chinchilla-Optimal Token Budget.** To test whether the benefits of CGLS persist at larger training scales, we replicate our main CGLS pretraining experiment on a 1B parameter model trained on 20B tokens, matching the Chinchilla-optimal parameter-to-data ratio (Hoffmann et al., 2022). At this scale, we observe in Table 2 that CGLS delivers clear gains over the randomized baseline across all benchmarks, and in perplexity on The Pile. Improvements are particularly notable on PIQA (+4.4 points), ARC-Easy (+5.2 points), and LAMBADA (+3.7 points), with the average accuracy rising from 38.4% to 42.3%. These results demonstrate that curriculum-guided layer scaling continues to confer benefits even as both model and dataset sizes grow, suggesting that its inductive bias remains useful at more realistic pretraining scales.

**Domain-Adaptive Code Pretraining.** Beyond scaling within a single corpus, we evaluate whether the coordinated data and model curriculum of CGLS can improve compute-efficiency in a domain-adaptive pretraining setting (Gururangan et al., 2020). Specifically, we instantiate this setting by evaluating CGLS under domain shift from general-purpose web text (2B tokens from DCLM) to code data (2B tokens of Python code from StarCoder2). Using HumanEval for downstream evaluation (Table 3), the baseline model trained on the randomized mixture of DCLM and StarCoder2 achieves modest pass@k performance.

In contrast, CGLS—trained on the same 4B tokens but ordered to progressively transition from general text to code—consistently outperforms the baseline in pass@k, e.g., nearly doubling pass@8 (6.7% vs. 3.6%). These results highlight that CGLS not only improves performance under fixed compute budgets, but also provides a structured mechanism for navigating distribution shifts between domains. More details on this experiment, including the data curriculum split, are provided in Appendix A.1.

## 5.2 ADDITIONAL ANALYSES

Beyond overall benchmark comparisons, we investigate two complementary aspects of CGLS. First, we assess stability through repeated training runs at the 2.5B-token scale (Section A.4), and through replacing the DistilBERT-based difficulty classifier with a FineWeb educational classifier (Penedo

| Model | Pile$\downarrow$ | PIQA$\uparrow$ | ARC-E$\uparrow$ | ARC-C$\uparrow$ | HS$\uparrow$ | LBD$\uparrow$ | Avg$\uparrow$ | GPQA$\uparrow$ | T9+$\uparrow$ | ORE$\uparrow$ | IB$\uparrow$ |
|---|---|---|---|---|---|---|---|---|---|---|---|
| | | | | Llama-3.2-1B (20B Tokens) | | | | | | | |
| BL-R | 26.33 | 63.00% | 41.79% | 23.38% | 33.52% | 30.37% | 38.41% | 25.22% | 2.787 | 17.30% | 0.235% |
| BL-C | 24.58 | 63.55% | 42.97% | 24.83% | 34.61% | 32.66% | 39.72% | 22.32% | 2.947 | 20.61% | 0.401% |
| CGLS | 22.99 | 67.36% | 46.97% | 26.28% | 36.80% | 34.08% | 42.30% | 29.24% | 3.568 | 17.03% | 0.539% |

Table 2: Evaluation results comparing CGLS and baseline approaches (BL-R refers to the compute-matched baseline trained on randomized data, and BL-C refers to the compute-matched baseline trained with a curriculum) at the 20B-token scale, aligned with the 20× data/parameter ratio in (Hoffmann et al., 2022). All models are compute-matched. The listed benchmarks are: perplexity on The Pile (Pile), PIQA (PIQA), ARC-Easy (ARC-E), ARC-Challenge (ARC-C), HellaSwag (HS), Lambada (LBD), the average over MC-QA tasks (Avg), GPQA (GPQA), TLDR9+ summarization (T9+), OpenRewriteEval (ORE), and the InfiniteBench long-context EN.QA task (IB). The best score for each metric is bolded, with the second-best method underlined.

| Model | pass@8 | pass@16 | pass@32 | pass@64 |
|---|---|---|---|---|
| | Llama-3.2-1B | | | |
| Baseline (Randomized) | 3.56% | 4.26% | 5.27% | 6.71% |
| CGLS | 6.67% | 7.63% | 8.78% | 9.76% |

Table 3: Results for a domain-shift experiment from general-purpose web documents (DCLM) to code data (StarCoder2), using HumanEval for downstream evaluation. As a baseline, we trained a 1B parameter model on a 50/50 mix of DCLM and StarCoder2 totaling 4B tokens. CGLS was trained on precisely the same data in a different order: a curriculum gradually transitioning from general text to code, and achieves higher pass@k scores for the full range of k considered.

et al., 2024) (Section A.5). Across three random seeds, CGLS achieves higher performance than the baseline across all benchmarks, indicating that the method's gains are robust to stochasticity in optimization. When using the FineWeb classifier, CGLS attains an average accuracy nearly identical to that obtained with the DistilBERT classifier, suggesting that the method is robust to the specific curriculum signal used to stratify data.

Second, we analyze the average validation accuracy across curriculum stages (Section A.7). These curves provide insight into how different training phases contribute to downstream performance. We observe steady gains for CGLS, particularly in later stages, whereas baseline methods plateau earlier. This progression underscores the role of synchronized model and data scaling in sustaining learning dynamics throughout training.

# 6 DISCUSSION

Inspired by how humans acquire cognitive capabilities in tandem with brain maturation and exposure to progressively more complex information, we align model capacity with data complexity during language model pretraining. We propose a new pretraining paradigm, Curriculum-Guided Layer Scaling (CGLS), that grows models in depth while adapting the data distribution to match their evolving capabilities. Applied to pretraining at GPT-2-Small and LLaMA-3.2-1B parameter counts, our approach yields consistent improvements in downstream task performance, with this trend holding at both the larger 1B-Chinchilla training scale and in a domain-adaptive code pretraining setting. In compute-matched comparisons, CGLS models outperform baselines on benchmarks requiring structured reasoning and domain-specific understanding, such as ARC and PIQA. Importantly, stacking-based baselines that expand model depth without synchronized data curricula do not improve downstream performance—CGLS significantly outperforms them across key benchmarks, demonstrating that aligning model growth with data complexity can unlock the benefits of progressive growing. Altogether, our results establish CGLS as a promising new paradigm for compute-efficient language model pretraining.

**Limitations.** While CGLS yields improvements across several knowledge-intensive and reasoning tasks, in a few tasks such as LAMBADA and HellaSwag, CGLS models perform comparably to the baselines, suggesting that not all tasks benefit equally from the progressive curriculum. We also observe that CGLS does not always lower perplexity: at the LLaMA-3.2-1B scale with 2.5B tokens, the randomized baseline achieved slightly better perplexity despite underperforming on reasoning tasks. Our intuition for this result is provided by studies of scaling laws with respect to downstream task performance Tay et al. (2022b;a), which find that some model inductive biases such as depth matter disproportionately more for downstream tasks than pretraining perplexity. CGLS is a curriculum over layer depth and therefore we postulate that there may be a related mechanism at play. This disconnect between perplexity and downstream performance highlights the need for better evaluation frameworks.

**Future Work.** Our experiments suggest that the initial model depth is a particularly important hyperparameter: starting with half the final number of layers ($N_1 = N/2$) yielded the strongest performance at both pretraining scales. Starting too small (e.g., $N_1 = 6$ at LLaMA-3.2-1B scale) led to weaker final models, likely due to insufficient capacity to capture basic linguistic patterns, while starting too large (e.g., $N_1 = 10$) provided no benefit and in some cases underperformed standard training (see Section A.3). This points to a potential sweet spot in progressive scaling where initial tractability and long-term transferability are best balanced. Additionally, our experiments with the FineWeb classifier suggest that curriculum construction is an important design choice, and that exploring improved or domain-adaptive difficulty classifiers may further strengthen CGLS. Broadly, these results highlight a rich design space for curriculum-guided scaling. Future work could explore (1) more finely tuning the stage lengths and dataset proportions, (2) exploring other notions—e.g., syntactic diversity, lexical richness, or factuality—for defining the curriculum, (3) applying curriculum-aware optimization schedules (e.g., varying learning rates with data complexity), (4) extending CGLS to larger parameter regimes (e.g. 7B), and (5) applying CGLS to non-text modalities (e.g., vision or multimodal data).

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

| GPT-2-Small (124M Parameters) | | |
|---|---|---|
| Stage | CGLS (Simple) | CGLS ($N_1 = 6$) |
| Stage 1 | 100 / 0 / 0 | 70 / 10 / 20 |
| Stage 2 | 0 / 100 / 0 | 15 / 25 / 60 |
| Stage 3 | 0 / 0 / 100 | 5 / 5 / 90 |
| New Layer Tuning | — | 45 / 30 / 25 |

Table 4: Data curricula per training stage for the three GPT-2-Small parameter scale setups: CGLS (Simple), CGLS (starting model depth $N_1 = 3$), and CGLS ($N_1 = 6$). Each split is represented as a proportion of TinyStories, BookCorpus, and DCLM tokens with format TinyStories / BookCorpus / DCLM, used for tuning only the new layers (New Layer Tuning) or for full tuning of the entire model at that stage. The Simple approach trains on one dataset for each stage, for both the tuning of only the new-layers and for fine-tuning, while the $N_1 = 3$ and $N_1 = 6$ approaches combine datasets with mixture weights that vary across stages.

| Llama-3.2-1B (1.2B Parameters) | | | |
|---|---|---|---|
| Stage | General (HS) | Domain (UG) | Specialized (Grad.) |
| Stage 1 | 25% | 50% | 25% |
| Stage 2 | 20% | 40% | 40% |
| Stage 3 | 15% | 35% | 50% |
| Stage 4 | 10% | 10% | 80% |
| New Layer Tuning | 34% | 33% | 33% |

Table 5: Data curricula per training stage for the Llama-3.2-1B parameter scale setups. The same set of optimized mixture weights are used for all CGLS experiments. Each stage is defined by mixture weights over three levels of complexity: high school–level (general understanding), undergraduate-level (some domain-specific knowledge), or graduate-level (highly specialized information).

Yang Zhang, Amr Mohamed, Hadi Abdine, Guokan Shang, and Michalis Vazirgiannis. Beyond Random Sampling: Efficient Language Model Pretraining via Curriculum Learning. 6 2025.

Wangbo Zhao, Yizeng Han, Jiasheng Tang, Kai Wang, Yibing Song, Gao Huang, Fan Wang, and Yang You. Dynamic Diffusion Transformer. 10 2024.

Yukun Zhu, Ryan Kiros, Richard Zemel, Ruslan Salakhutdinov, and others. Aligning Books and Movies: Towards Story-like Visual Explanations by Watching Movies and Reading Books. 6 2015.

Barret Zoph, Irwan Bello, Sameer Kumar, Nan Du, Yanping Huang, Jeff Dean, Noam M Shazeer, and William Fedus. ST-MoE: Designing Stable and Transferable Sparse Expert Models. 2022. URL https://api.semanticscholar.org/CorpusID:248496391.

# A APPENDIX

## A.1 EXPERIMENTAL SETUP

All training was performed using the Hugging Face Transformers library (Wolf et al., 2019) on NVIDIA H100 80GB GPUs with NVLink. Each model was initialized with a learning rate of $2 \times 10^{-4}$ and optimized using AdamW with a weight decay of 0.01. A warmup-stable-decay (WSD) learning rate schedule was applied, with 1,000 warmup steps during the initial stage (smallest model), followed by a stable learning rate across intermediate stages, and cosine decay to zero during the final stage (largest model). Layers added after model expansion were initially trained with a higher learning rate of $5 \times 10^{-4}$, for 20% of the compute budget for the given stage. Optimizer states were reset between stages for all experiments involving layer expansion.

| Model | # Shots | PIQA | ARC-E | ARC-C | HellaSwag | LAMBADA | Average |
|---|---|---|---|---|---|---|---|
| **Llama-3.2-1B w/ 2.5B Tokens** | | | | | | | |
| Baseline (Randomized) | 0 | 54.41% | 32.69% | 23.98% | 31.41% | 29.89% | 34.48% |
| | 5 | 56.31% | 33.15% | 26.19% | 31.32% | 31.32% | 35.66% |
| Baseline (Curricularized) | 0 | 53.86% | 27.30% | 30.19% | 30.19% | 34.02% | 35.11% |
| | 5 | 54.73% | 29.27% | 29.73% | 29.73% | 33.57% | 35.41% |
| Layer-Scaling Only | 0 | 53.54% | 26.71% | 28.89% | 28.46% | 27.67% | 33.05% |
| | 5 | 54.95% | 26.71% | 29.38% | 29.11% | 28.37% | 33.70% |
| MIDAS | 0 | 51.58% | 26.54% | 30.19% | 28.89% | 25.91% | 32.62% |
| | 5 | 57.18% | 28.33% | 31.81% | 29.38% | 29.42% | 35.22% |
| CGLS | 0 | 56.69% | 33.75% | 29.18% | 31.95% | 31.04% | **36.52%** |
| | 5 | 58.65% | 33.65% | 29.10% | 32.02% | 31.95% | **37.07%** |

Table 6: Zero- and 5-shot evaluation results across benchmarks for different training strategies. CGLS achieves the best average at both shot settings (bold).

The full stage-wise splits of TinyStories, BookCorpus, and DCLM for the GPT-2-Small experiments are provided in Table 4. For Llama-3.2-1B, the stratified mixture weights over general, domain-specific, and specialized subsets of DCLM are in Table 5. For the domain shift experiment from DCLM (general web text) to StarCoder (Python code), we used a four-stage progression with mixture weights of 80/20, 70/30, 60/40, and 50/50 (DCLM/StarCoder), with all other hyperparameters identical to the other experiments.

Training a 1B parameter model with CGLS on 2.5B tokens took 5h53m on 4x H100s, while the compute-matched baseline took 5h39m on the same configuration. We note that our implementation of CGLS is not yet optimized for efficiency, e.g., with checkpoint loading from a fast disk. Moreover, no aspects of CGLS should meaningfully decrease model FLOPs utilization (MFUs), and CGLS and the baseline are compute-matched. Given this, we anticipate the wall-clock time of CGLS and the baseline should be essentially identical.

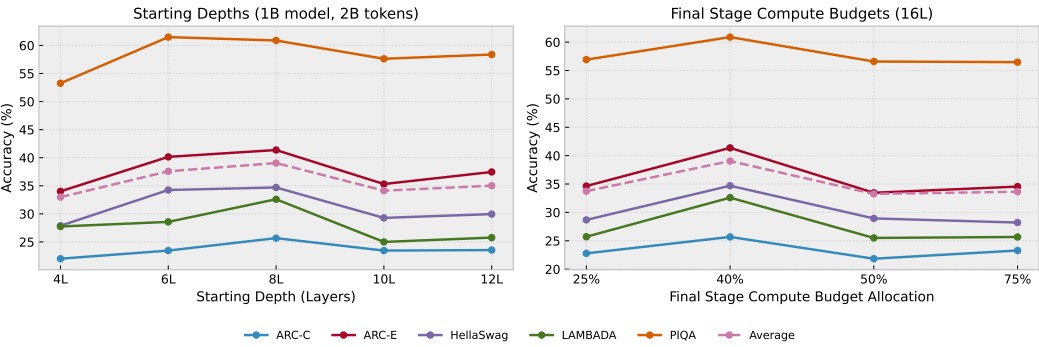

Figure 2: Results for 1D hyperparameter sweeps over the starting model depth and stage-wise training budgets for the 1B-parameter model, trained on 2.5B tokens from DCLM. The average across the benchmarks is denoted with a dashed line.

## A.2 FEW SHOT RESULTS

In standard academic pretraining works, few-shot evaluations are uncommon because effective in-context learning emerges at larger scales, as shown in the GPT-3 paper Brown et al. (2020). For example, recent large-scale academic works such as H-Nets Hwang et al. (2025), which pretrains 1B parameter models for 100B tokens, do not include few-shot evaluations. Nevertheless, we conduct some few-shot evaluations, with PMI accuracy Gu et al. (2024) to reduce variance, and report the

| Model | PIQA | ARC-E | ARC-C | HellaSwag | Lambda | Average |
|---|---|---|---|---|---|---|
| **Llama-3.2-1B w/ 2.5B Tokens** | | | | | | |
| Baseline | $59.18 \pm 0.28\%$ | $36.46 \pm 0.42\%$ | $22.38 \pm 0.78\%$ | $27.81 \pm 0.24\%$ | $20.49 \pm 0.36\%$ | $33.26 \pm 0.24\%$ |
| CGLS | $\mathbf{62.19 \pm 1.42\%}$ | $\mathbf{40.47 \pm 2.43\%}$ | $\mathbf{23.78 \pm 1.76\%}$ | $\mathbf{30.06 \pm 1.24\%}$ | $\mathbf{23.22 \pm 1.19\%}$ | $\mathbf{35.94 \pm 1.60\%}$ |

Table 7: Experimental replication results comparing CGLS and baseline approaches at the 2.5B token scale. Both models are compute-matched with training runs repeated three times in each case. We report zero-shot results with standard errors, with CGLS achieving a higher score across all benchmarks.

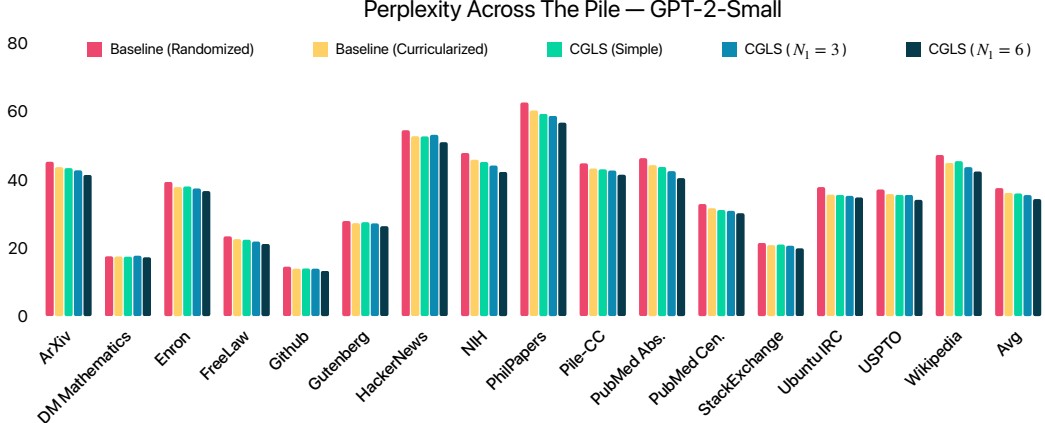

Figure 3: Model perplexity at GPT-2-Small parameter count across various individual subsets of The Pile, for three CGLS configurations and the strongest two baselines. Lower perplexity scores indicate better performance. CGLS consistently outperforms the baselines, particularly on complex knowledge-intensive datasets such as ArXiv, PubMed, and NIH.

results in Table 6. For both zero- and 5-shot evaluation, CGLS achieves the best average across the benchmarks.

### A.3 HYPERPARAMETER ABLATIONS

We conduct ablations of core hyperparameters such as stage-wise training budgets and the starting depth. Though we are not able to exhaustively tune these hyperparameters on our academic compute budget, we report below some results conducting 1D sweeps on these key parameters. We find in figure 2 that starting at half the final depth ($N_1 = 8$) consistently yields the best results across downstream tasks, and that allocating 40% of the compute budget to the final stage performs best, outperforming front-heavy (25%) and back-heavy (50%, 75%) schedules. Increasing the starting depth beyond half the final depth did not yield additional improvements. These findings suggest that a balanced schedule that gradually expands capacity optimizes performance. Additionally, starting from half the model depth may strike a near-optimal balance between early-stage tractability and representational richness in progressive training frameworks.

### A.4 STABILITY

Replication experiments at the 1B scale (Table 7) confirm that the improvements from CGLS are consistent across multiple seeds. While the variance across runs is slightly larger for CGLS than for the baseline, the mean accuracy is higher on every benchmark, indicating that the method's benefits are robust. This suggests that progressive scaling may introduce some additional variability in optimization dynamics, but reliably yields stronger final models.

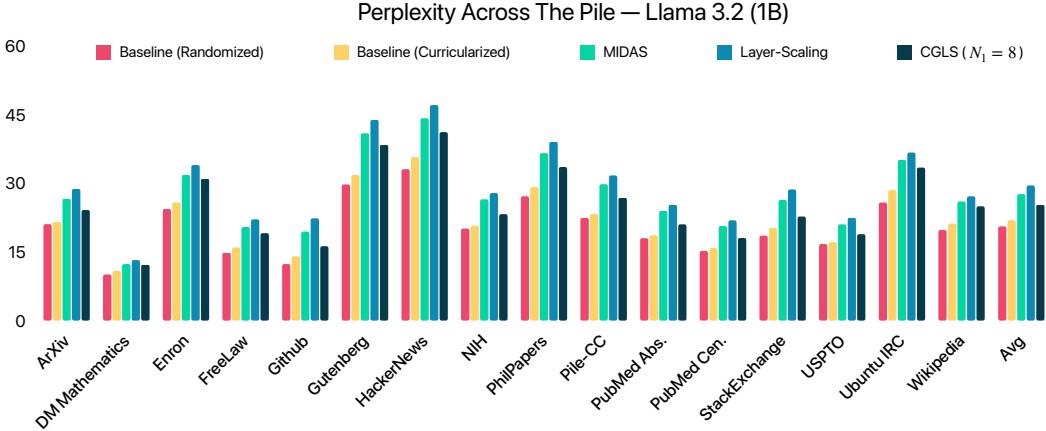

Figure 4: Model perplexity across various individual subsets of The Pile for all baselines versus CGLS for Llama-3.2-1B. Lower perplexity scores indicate better performance. Across all subsets, the randomized baseline achieves the lowest perplexity, with progressive scaling models outperforming MIDAS and layer scaling without a data curriculum.

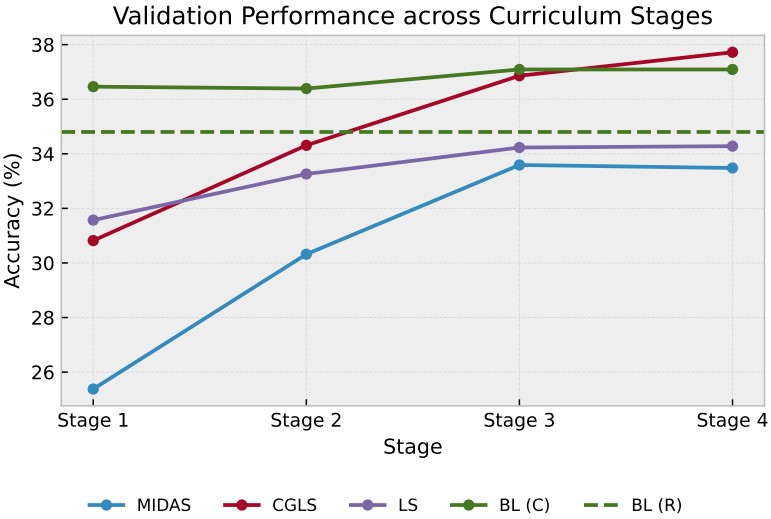

Figure 5: Validation performance at each training stage for all methods. BL refers to the baseline, with R denoting the randomized variant, and C the curricularized variant.

## A.5 ROBUSTNESS TO THE DIFFICULTY CLASSIFIER

A core component of CGLS is the curriculum used to order training data by difficulty. In our primary experiments, difficulty stratification is performed using a DistilBERT-based classifier trained to predict coarse complexity labels. Since model performance may depend on the quality or structure of this classifier, we assess the robustness of CGLS to alternate curriculum signals.

To this end, we conducted an additional experiment in which we replaced the DistilBERT-based classifier with the FineWeb educational difficulty classifier (Penedo et al., 2024), which provides a substantially different distributional prior and difficulty spectrum. Table 8 reports results for the baseline, the original CGLS configuration, and the variant that uses FineWeb for curriculum construction.

| Model | PIQA$^\uparrow$ | ARC-E$^\uparrow$ | ARC-C$^\uparrow$ | HellaSwag$^\uparrow$ | Lambda$^\uparrow$ | Average$^\uparrow$ |
|---|---|---|---|---|---|---|
| **Llama-3.2-1B w/ 2.5B Tokens** | | | | | | |
| Baseline | 59.09 | 36.36 | 24.24 | 34.20 | 32.99 | 37.38 |
| CGLS | 61.21 | 41.37 | 25.43 | **34.65** | **32.74** | **39.08** |
| FineWeb | **61.59** | **45.20** | **26.37** | 31.30 | 30.84 | 39.06 |

Table 8: Comparison of CGLS using the default DistilBERT-based difficulty classifier vs. a FineWeb-based difficulty classifier. All benchmarks are conducted in a zero-shot setting. FineWeb yields similar average performance to standard CGLS, with stronger results on ARC-E and ARC-C but lower performance on HellaSwag and Lambada.

Across benchmarks, the FineWeb-based CGLS variant performs comparably to the DistilBERT-based version: both substantially outperform the baseline, and their average accuracies (39.06% vs. 39.08%) are nearly identical. The FineWeb classifier produces modest improvements on PIQA, ARC-Easy, and ARC-Challenge, while slightly reducing performance on HellaSwag and Lambada. These shifts reflect differences in how the two classifiers stratify training data, but the overall stability indicates that CGLS is not overly sensitive to the specific classifier used to define difficulty.

These findings suggest that CGLS benefits from the existence of a meaningful progression in data complexity, rather than from the idiosyncrasies of a particular classifier. We therefore view improved, domain-adaptive, or multi-signal difficulty estimators as an important direction for future work, and emphasize that CGLS is compatible with a wide family of curriculum-generation strategies beyond DistilBERT.

## A.6 PERPLEXITY RESULTS

At the GPT-2-Small scale, CGLS achieves lower perplexity than the randomized and curricularized baselines on average, but the gains are most pronounced on challenging, knowledge-intensive subsets of The Pile such as *PubMed Abstracts*, *PhilPapers*, and *Wikipedia* (Figure 3).

For Llama-3.2-1B trained on 2.5B tokens, however, we observe the opposite trend: CGLS yields higher perplexity than the baselines across most subsets of The Pile (Figure 4). As discussed in the main paper, this divergence highlights that perplexity does not always align with downstream task improvements, particularly at smaller training budgets. Notably, scaling to a Chinchilla-optimal 20B tokens, this discrepancy disappears, and CGLS improves both perplexity and downstream accuracy.

## A.7 PERFORMANCE BY STAGE

Figure 5 reports the average accuracy across the benchmarks after each stage of training for CGLS and competing methods. The results highlight two main trends. First, CGLS improves steadily across stages, surpassing the randomized baseline by Stage 3 and continuing to gain performance in the final expansion. Second, while alternative progressive strategies such as MIDAS and LS also show early improvements, their performance plateaus sooner, converging below the baseline reference. In contrast, CGLS maintains upward momentum throughout the curriculum, suggesting that its synchronized data and model growth more effectively transfers knowledge from earlier stages into the final model.

