# OpenReview forum: "Curriculum-Guided Layer Scaling for Language Model Pretraining"
_ICLR.cc/2026/Conference — Submitted to ICLR 2026_

### Official Review · Reviewer_t3pK · 2025-10-31

**Soundness:** 3
**Presentation:** 3
**Contribution:** 3
**Rating:** 4
**Confidence:** 4

**Summary:**

The authors propose the use of layer scaling combined with curriculum learning to improve the efficiency of LLM pretraining. Layer scaling progressive increases the model size, while curriculum learning progressively increases the difficulties of the samples. Their empirical results show that the combination outperforms naive layer scaling.

**Strengths:**

1. The research direction of progressively increasing the model size is interesting and worth exploring.

**Weaknesses:**

1. Overall, the novelty and insight provided by this paper is limited. Various settings used in the experiments seem arbitrary which leads to concern about scalability.
2. The comparison and the improvement over the baseline seem not to be very solid. The setup for the curriculum learning baseline is not clearly specified, while there are various stronger baselines in this category in the literature. For 700M and 2.5B token setting, the improvement over the curriculum learning baseline is slight, while it is unclear how much tuning is conducted for each of the methods. For 20B token setting, comparison with curriculum learning baseline  is not shown.

**Questions:**

1. The combination of layer scaling and curriculum learning also increases the design complexity. How sensitive are the proposed methods to these settings (When and how many layers are scaled at a time / How large and how much training each curriculum be)?
2. Do you have results on larger models? This seems to be important as various settings in the experiment seem arbitrary which leads to concern about scalability.

---

> ### Author Response · Authors · 2025-11-25
> **Response to Reviewer t3pK**
>
> We thank the reviewer for their helpful comments and noting that the research direction of progressively increasing model size is “interesting and worth exploring.” In the following response, we clarify the design motivations, address questions regarding hyperparameter choices and scalability, and provide new analyses, including a curriculum baseline at the 20B-token scale, classifier-robustness experiments, and expanded evaluations, that strengthen the empirical grounding of CGLS.
>
> ### Comment 1
>
> > Overall, the novelty and insight provided by this paper is limited. Various settings used in the experiments seem arbitrary which leads to concern about scalability.
>
> Thank you for the feedback. We have clarified several aspects of the method and experimental design to better highlight the contribution and provide additional evidence of robustness.
>
> First, CGLS introduces a coordinated training strategy in which model depth and data difficulty progress together. In contrast, prior work varies only one of these dimensions at a time (e.g., curriculum learning with a fixed model or layer expansion without a data curriculum). This joint treatment provides a distinct training trajectory that, as shown in our experiments, leads to consistent improvements across scales, domains, and compute budgets.
>
> To address concerns about arbitrary parameters, we refer the reviewer to Sections 4.2 and Appendix D, which detail the selection of hyperparameters and include systematic sensitivity analyses. These include 1D sweeps over the starting depth N_1 and the final-stage compute budget. The results show stable trends with clear optima (e.g., N_1 = N/2), indicating that the method’s performance is not sensitive to fragile or highly tuned settings.
>
> Regarding scalability, we evaluate CGLS not only at the 1B-parameter level but also at a 20B-token Chinchilla-optimal run, and under a domain-shift setting. These larger-scale experiments show that the method remains effective as the training budget increases, and they help contextualize the smaller-scale results. Current academic compute budgets make fully compute-matched experiments at 7B+ scale infeasible. Our choice of the 1B parameter regime is consistent with recent academic pretraining works that use 0.5B–1.3B models as scalable, cost-effective proxies for studying new pretraining algorithms [1,2]. Given these points, we view scaling CGLS to 7B–70B as exciting future work, but not feasible in an academic setting. We will clarify this motivation in the revision.
>
> [1] Qi et al., “EvoLM: In Search of Lost Language Model Training Dynamics,” 2025, arXiv:2506.16029.
> [2] Hwang et al., “Dynamic Chunking for End-to-End Hierarchical Sequence Modeling,” 2025, arXiv:2507.0795.
>
> ### Comment 2
>
> > The comparison and the improvement over the baseline seem not to be very solid. The setup for the curriculum learning baseline is not clearly specified, while there are various stronger baselines in this category in the literature. For 700M and 2.5B token setting, the improvement over the curriculum learning baseline is slight, while it is unclear how much tuning is conducted for each of the methods. For 20B token setting, comparison with curriculum learning baseline is not shown.
>
> Thank you for raising this point. We have clarified the definition of the curriculum baseline in Section 4.2. Both the baseline curriculum and CGLS use the same difficulty classifier, the same three difficulty tiers, and the same pacing schedule; all optimization hyperparameters and compute budgets are matched. The only difference is whether the model depth is expanded during training.
>
> We also examined stronger curriculum-learning baselines from the literature, particularly Beyond Random Sampling (BRS) [1]. This work evaluates several curriculum variants with 10 difficulty groups and finds that 3-group curricula often perform comparably or only slightly worse. Since our goal is to isolate the effect of synchronizing data complexity with model growth, we adopt the simpler 3-tier curriculum as a clean control condition. We have added a discussion about BRS to the Related Works section.
>
> To address the absence of a curriculum baseline at the 20B-token scale, we have added new experiments in the revision. The 20B-token curriculum baseline performs between the random baseline and CGLS across all reported metrics, confirming that the relative ordering observed at more minor scales also holds at a larger budget. We additionally added a variety of new benchmarks for this 20B-token experiment scale, including GPQA (reasoning), InfiniteBench (long-context reasoning), TLDR9+ (summarization), and OpenRewriteEvaluation, across all of which CGLS performs equally or better than the baselines.
>
> [1] Zhang et al., “Beyond Random Sampling: Efficient Language Model Pretraining via Curriculum Learning,” arXiv:2506.11300.
>
> [Continued in next comment...]

---

> > ### Author Response · Authors · 2025-11-25
> > **Response to Reviewer t3pK (continued)**
> >
> > ### Comment 3
> >
> > > The combination of layer scaling and curriculum learning also increases the design complexity. How sensitive are the proposed methods to these settings (When and how many layers are scaled at a time / How large and how much training each curriculum be)?
> >
> > We agree that combining layer scaling with curriculum learning introduces additional design choices, and we address this through explicit sensitivity analyses. We point the reviewer to appendices A.3-A.5, which include:
> >
> > * A sweep over the starting depth N_1 \in \{4, 6, 8, 10, 12\}
> > * A sweep over final-stage compute allocation (25%, 40%, 50%, 75%)
> > * Stability analysis of CGLS with multiple replications.
> > * Robustness analysis to the difficulty classifier or choice of curriculum.
> >
> > These experiments reveal clear trends: N_1 = N/2 and mid-range stage weights consistently yield competitive results. We point to these results in sections 4.2 and 5.2.
> >
> > ### Comment 4
> >
> > > Do you have results on larger models? This seems to be important as various settings in the experiment seem arbitrary, which leads to concern about scalability.
> >
> > Thank you for raising this point. We added clarification to the paper on why 1B-scale models were selected. The 0.7B–1.3B range is widely used in recent academic pretraining studies as a practical and computationally accessible setting for studying training algorithms and curricula [1,2]. Examples include works on data mixtures, difficulty-aware sampling, inference-efficient pretraining, and progressive depth training, many of which use 1B-scale models as their primary testbed due to their affordability and ability to run multiple controlled ablations.
> >
> > Running compute-matched experiments at 7B+ scale with a reasonable token count is beyond the scope of academic compute resources. Instead, we provide scalability evidence through (1) a 20B-token Chinchilla-optimal experiment, and (2) a domain-shift experiment. Both confirm that the method remains effective when the training budget increases, and even when the task changes during training.
> >
> > [1] Qi et al., “EvoLM: In Search of Lost Language Model Training Dynamics,” 2025, arXiv:2506.16029.
> >
> > [2] Hwang et al., “Dynamic Chunking for End-to-End Hierarchical Sequence Modeling,” 2025, arXiv:2507.0795.

---

### Official Review · Reviewer_c3To · 2025-11-01

**Soundness:** 3
**Presentation:** 3
**Contribution:** 2
**Rating:** 4
**Confidence:** 4

**Summary:**

This paper proposes Curriculum-Guided Layer Scaling (CGLS), a novel pretraining framework for language models that synchronizes progressive model growth (layer stacking) with a data curriculum of increasing complexity. The core idea is inspired by cognitive development and aims to improve computational efficiency and downstream performance. The authors conduct experiments at multiple scales (100M and 1.2B parameters) and show that CGLS outperforms several baselines, including standard training, curriculum-only, and layer-scaling-only approaches, on various reasoning and knowledge-intensive benchmarks.

**Strengths:**

1. The core design of the paper is quite interesting and can provide some inspiration for the field.

2. Write clearly and easily understandable.

**Weaknesses:**

1. The downstream evaluation focuses heavily on multiple-choice QA tasks (PIQA, ARC). While these are standard, they don't fully capture the breadth of LLM capabilities. The evaluation would be strengthened by including math benchmarks (e.g., GSM8K, AIME 2024, AIME 2025…), and coding benchmarks (e.g., Humaneval…).

2 . The experiments only reach 1B parameters and are confined to the Llama architecture, failing to demonstrate scalability to modern 7B+ models or generalization across diverse architectures (e.g., Qwen…l).

3. No evaluation on dedicated reasoning benchmarks and basic reasoning models is provided.

**Questions:**

Please see the weaknesses.

---

> ### Author Response · Authors · 2025-11-25
> **Response to Reviewer c3To**
>
> We thank the reviewer for the constructive feedback and for highlighting that the core design of CGLS is “quite interesting” and potentially inspiring for the field. We also appreciate the comments on the clarity and accessibility of the paper. Below, we address the reviewer’s concerns regarding evaluation breadth, baseline comparisons, and scalability, and we provide several new experiments—including GPQA, summarization, long-context reasoning, and domain-shift results—that broaden and reinforce the empirical support for our method.
>
> ### Comments 1 and 3
>
> > The downstream evaluation focuses heavily on multiple-choice QA tasks (PIQA, ARC). While these are standard, they don't fully capture the breadth of LLM capabilities. The evaluation would be strengthened by including math benchmarks (e.g., GSM8K, AIME 2024, AIME 2025…), and coding benchmarks (e.g., Humaneval…).
>
> > No evaluation on dedicated reasoning benchmarks and basic reasoning models is provided.
>
> We appreciate these suggestions. While our main experiments focus on standard multiple-choice QA benchmarks (PIQA, ARC), we agree that it is valuable to evaluate a broader capability range. **We already include coding performance on HumanEval in Table 3**, and in the revision, we now additionally report:
>
> * **Math reasoning** (GSM8K 8-shot CoT, AIME 2024, and MATH zero-shot): As expected for academic-scale pretraining experiments [1], both the baseline and CGLS perform near-chance (≈1%).
> * **GPQA** (main, zero-shot reasoning): CGLS (20B) achieves 29.24 ± 2.15%, outperforming both the compute-matched randomized baseline (25.22 ± 2.05%) and the curricularized baseline (22.32 ± 1.97%).
> * **Long-context reasoning** (InfiniteBench): all models achieve relatively low scores, with CGLS (20B)’s F1 of 0.539 ± 0.150% being the strongest, compared to 0.235 ± 0.036% for the randomized baseline and 0.401 ± 0.117% for the curricularized baseline.
> * **Summarization** (TLDR9+): CGLS (20B) achieves the highest rougeL at 3.568 ± 0.159, compared to 2.787 ± 0.143 (randomized) and 2.9470 ± 0.148 (curricularized) for the baselines.
> * **Open-rewrite evaluation**: CGLS performs on par with the randomized baseline (17.03 ± 0.29% vs. 17.30 ± 0.29%), while the curricularized baseline is moderately higher (20.61 ± 0.31%). We note that, unlike the other evaluations, this task is more sensitive to stylistic factors than to underlying model quality. Importantly, CGLS does not degrade performance in this setting.
>
> Across these expanded evaluations, CGLS maintains competitive or superior performance while not harming general-purpose abilities. **Notably, on GPQA, a high-difficulty reasoning benchmark that probes graduate-level knowledge, CGLS outperforms the best compute-matched baseline by over 4%** — we believe this is a substantial improvement for this model scale.  We incorporated these new results into the revision in Table 2.
>
> [1] Wei et al., “Emergent Abilities of Large Language Models,” 2022, arXiv: 2206.07682.
>
> ### Comment 2
>
> > The experiments only reach 1B parameters and are confined to the Llama architecture, failing to demonstrate scalability to modern 7B+ models or generalization across diverse architectures (e.g., Qwen…l).
>
> We agree that testing on larger models would be ideal. However, current academic compute budgets make fully compute-matched experiments at 7B+ scale infeasible. Our choice of the 1B parameter regime is consistent with recent academic pretraining works that use 0.5B–1.3B models as scalable, cost-effective proxies for studying new pretraining algorithms [1,2]. These sizes allow us to perform controlled ablations (stage depth, data curricula, stage lengths) while fully matching FLOPs across methods.
>
> Importantly, we also evaluate CGLS at a Chinchilla-optimal 20B-token budget, demonstrating that the method remains effective when substantially scaling up data. We view extending CGLS to 7B–70B parameters as exciting future work, but it is not feasible given our academic compute budget. We clarified this motivation in the revision.
>
> [1] Qi et al., “EvoLM: In Search of Lost Language Model Training Dynamics,” 2025, arXiv:2506.16029.
>
> [2] Hwang et al., “Dynamic Chunking for End-to-End Hierarchical Sequence Modeling,” 2025, arXiv:2507.0795.

---

### Official Review · Reviewer_xpnx · 2025-11-01

**Soundness:** 3
**Presentation:** 3
**Contribution:** 3
**Rating:** 6
**Confidence:** 3

**Summary:**

The paper introduces Curriculum-Guided Layer Scaling (CGLS), a pretraining paradigm for large language models (LLMs) that synchronizes model growth (via progressive layer stacking) with data curriculum learning. Specifically, at each training stage, several new transformer layers are added and trained in isolation while earlier layers are frozen. The entire model is then unfrozen and fine-tuned on a dataset of higher complexity, with data difficulty measured using a DistilBERT-based classifier or intrinsic dataset structure.

The authors conduct experiments at multiple scales: GPT-2-Small (124M parameters), LLaMA-3.2-1B (1.2B parameters), and a Chinchilla-optimal 1B–20B token setup, as well as a domain-shift setting from general text to code. Across these settings, CGLS consistently outperforms compute-matched baselines, including progressive stacking (MIDAS), layer-scaling-only, and curriculum-only methods. The approach yields improved reasoning and generalization performance on benchmarks such as PIQA, ARC-Easy, and HellaSwag.

**Strengths:**

1. The integration of curriculum learning and progressive layer scaling into a unified pretraining framework is novel for LLM pretraining. The approach is conceptually sound and aligns well with cognitive principles of gradual learning.

2. The experiments are thorough and well-validated across multiple model scales, with replication and ablation studies supporting the claims. Results show consistent and meaningful gains on reasoning-focused benchmarks such as PIQA and ARC, demonstrating improved efficiency without additional computational cost.

3. The paper is clearly written, well-organized, and easy to understand.

**Weaknesses:**

1. Most experiments focus on reasoning and question-answering benchmarks. Broader capabilities such as dialogue generation, summarization, or open-ended tasks are not evaluated, leaving the generality of the approach less explored.
2. The effectiveness of CGLS relies on the accuracy of the DistilBERT-based difficulty classifier for data stratification, which may not generalize well to other languages, domains, or modalities.
3. The method is only tested on text-based pretraining; no experiments are conducted on multimodal or speech data, which limits evidence of its broader applicability.

**Questions:**

Please refer to the weaknesses section.

---

> ### Author Response · Authors · 2025-11-25
> **Response to Reviewer xpnx**
>
> We thank the reviewer for the thoughtful and positive assessment of our work. We are glad that you found the integration of curriculum learning with progressive layer scaling to be "novel, conceptually sound, and well-aligned with cognitive principles". We also appreciate your recognition of the clarity of the writing and the thoroughness of the experimental validation across multiple scales and settings. We address your concerns in detail below and present additional experiments, including expanded benchmarks and classifier-robustness analysis, that further strengthen the contributions of the paper.
>
> ### Comment 1
>
> > Most experiments focus on reasoning and question-answering benchmarks. Broader capabilities such as dialogue generation, summarization, or open-ended tasks are not evaluated, leaving the generality of the approach less explored.
>
> We thank the reviewer for the suggestion to broaden the evaluation beyond multiple-choice QA. We have now added several additional benchmarks for our largest experiment that probe different model capabilities:
>
> * **Math reasoning** (GSM8K 8-shot CoT, AIME 2024, and MATH zero-shot): As expected for academic-scale pretraining experiments [1], both the baseline and CGLS perform near-chance (≈1%).
> * **GPQA** (zero-shot reasoning): CGLS (20B) achieves 29.24 ± 2.15%, outperforming both the compute-matched randomized baseline (25.22 ± 2.05%) and the curricularized baseline (22.32 ± 1.97%).
> * **Long-context reasoning** (InfiniteBench): all models achieve relatively low scores, with CGLS (20B)’s F1 of 0.539 ± 0.150% being the strongest, compared to 0.235 ± 0.036% for the randomized baseline and 0.401 ± 0.117% for the curricularized baseline.
> * **Summarization** (TLDR9+): CGLS (20B) achieves the highest rougeL at 3.568 ± 0.159, compared to 2.787 ± 0.143 (randomized) and 2.9470 ± 0.148 (curricularized) for the baselines.
> * **Open-rewrite evaluation**: CGLS performs on par with the randomized baseline (17.03 ± 0.29% vs. 17.30 ± 0.29%), while the curricularized baseline is moderately higher (20.61 ± 0.31%). We note that, unlike the other evaluations, this task is more sensitive to stylistic factors than to underlying model quality. Importantly, CGLS does not degrade performance in this setting.
>
> Across these expanded evaluations, CGLS maintains competitive or superior performance while not harming general-purpose abilities. We have incorporated these new results into the revision in Table 2.
>
> [1] Wei et al., “Emergent Abilities of Large Language Models,” 2022, arXiv: 2206.07682.
>
> ### Comment 2
>
> > The effectiveness of CGLS relies on the accuracy of the DistilBERT-based difficulty classifier for data stratification, which may not generalize well to other languages, domains, or modalities.
>
> Thank you for raising this point. We agree that CGLS performance depends on curriculum construction and, therefore, on the accuracy of the difficulty classifier used to stratify the training data in our approach. To investigate this, we conducted an additional experiment in which we replaced our DistilBERT-based difficulty classifier with the FineWeb educational classifier [1], which provides a substantially different distribution and difficulty signal. The results of this comparison are shown below:
>
> | Model    | PIQA  | ARC-E | ARC-C | HellaSwag | Lambada | Average |
> |----------|-------|-------|-------|-----------|---------|---------|
> | Baseline | 59.09 | 36.36 | 24.24 | 34.20     | 32.99   | 37.38   |
> | CGLS     | 61.21 | 41.37 | 25.43 | 34.65     | 32.74   | 39.08   |
> | FineWeb  | 61.59 | 45.20 | 26.37 | 31.30     | 30.84   | 39.06   |
>
> We find that using the FineWeb classifier achieves performance comparable to CGLS in the DistilBERT-based setting: performance improves over the baseline across most benchmarks, and the average accuracy is nearly identical to the original CGLS configuration. Notably, the FineWeb classifier yields slightly higher scores on PIQA, ARC-Easy, and ARC-Challenge, while modestly reducing performance on HellaSwag and Lambada. This suggests that CGLS is not overly sensitive to the specific classifier used, and that multiple curriculum signals can support similar overall gains. We highlight in the discussion that improved or domain-adaptive curriculum classifiers are an important direction for future work, and that CGLS is compatible with alternative complexity measures or methods of constructing curricula beyond DistilBERT.
>
> [1] Marion et al., “When Less is More: Investigating Data Pruning for Pretraining LLMs at Scale,” 2023, arXiv: 2309.04564.
>
> [Continued in next comment...]

---

> > ### Author Response · Authors · 2025-11-25
> > **Response to Reviewer xpnx (continued)**
> >
> > ### Comment 3
> >
> > > The method is only tested on text-based pretraining; no experiments are conducted on multimodal or speech data, which limits evidence of its broader applicability.
> >
> > We appreciate the reviewer’s interest in the broader applicability of CGLS beyond text. While our method is conceptually modality-agnostic, we focus this work specifically on text pretraining for two reasons:
> >
> > Scope: CGLS introduces an additional dimension in the pretraining design space, and evaluating it across multiple modalities would make it difficult to isolate its core effects. Text is the largest and most stable experimental setting for controlled ablations under fixed compute budgets.
> >
> > Established baselines: Most closely related approaches (e.g., progressive stacking, difficulty-aware curricula, depth-scaling baselines) are developed in the context of language models. Staying in this domain allows for direct, compute-matched comparisons and avoids confounding factors from multimodal architectures.
> >
> > We agree that extending CGLS to vision and multimodal models is an exciting next step—especially for domains where data complexity increases more sharply (e.g., video, medical, or instruction-tuned multimodal tasks). We clarified this point more explicitly in the Future Work section.

---

### Author Response · Authors · 2025-11-25
**Summary of Rebuttal and Revisions**

We thank all reviewers for their thoughtful feedback and for highlighting the strengths of our work, including that it is “novel… conceptually sound and aligned with cognitive principles of gradual learning” [xpnx], “well-written and easily understandable” [c3To], and that “the research direction of progressively increasing the model size is interesting and worth exploring” [t3pK]. We appreciate these encouraging assessments and have expanded our experiments and clarifications accordingly.

Below, we summarize key additions and improvements made during the rebuttal period.

1) Broader Evaluation: reviewers encouraged a more diverse evaluation suite. We have now added **four** new benchmark families at the 1B/20B-token scale: GPQA, where CGLS outperforms the best baseline **by over 4%**; InfiniteBench, a long-context reasoning task where CGLS achieves the strongest F1 score (0.539 vs. 0.235 and 0.401 for the baselines); TLDR9+, a summarization task where CGLS obtains the highest rougeL (3.568 vs. 2.787 / 2.947), and OpenRewriteEval, where CGLS performs on par with the randomized baseline (17.03% vs. 17.30%). These results strengthen the generality of our method beyond multiple-choice QA.

2) Additional Curriculum-Classifier Experiment: A reviewer raised concerns about reliance on the DistilBERT difficulty classifier. To address this, we trained a FineWeb-based classifier and re-ran the CGLS experiment at the 1B parameter and 2.5B token scale. Notably, the average score across the tested benchmarks was nearly identical with either the FineWeb classifier (39.06%) or DistilBERT as presented in the paper (39.08%). These near-identical averages and improved scores on three benchmarks indicate that CGLS is not overly sensitive to the difficulty classifier, and that multiple curriculum signals can yield similar benefits for pretraining.

3) Curricularized Baseline at the 20B-Token Scale: We additionally implemented a curricularized baseline at the 20B-token scale, which performs between the random baseline and CGLS, matching the trends seen at the other tested scales.

We are grateful to the reviewers for their constructive feedback and for recognizing the paper’s clarity, conceptual motivation, and empirical validation. The newly added experiments: math reasoning, GPQA, long-context tasks, summarization, open-ended rewriting, classifier sensitivity tests, and a 20B-token curriculum baseline, substantially strengthen the empirical foundation of CGLS.
We hope these additions address all concerns and reinforce the core message that **coordinating model growth with a data curriculum provides a simple, compute-efficient, and robust improvement to language model pretraining.**

---

### Meta-Review · Area_Chair_6rG7 · 2026-01-12

**Summary:**

This paper proposes Curriculum-Guided Layer Scaling (CGLS), a pretraining framework that jointly schedules model growth—via progressive layer stacking—and data curriculum learning, where increasingly complex data is introduced as the model expands. Experiments across multiple scales (GPT-2-Small, LLaMA-3.2-1B) and domains (including code) show consistent improvements over compute-matched baselines on reasoning benchmarks like PIQA, ARC-Easy, and HellaSwag.

However, the work has several limitations. The novelty is incremental, combining existing ideas of curriculum learning and progressive layer stacking without deep theoretical or mechanistic insight. Experimental design raises concerns: several choices (e.g., data difficulty metrics, stage-wise layer counts) appear arbitrary, and comparisons with curriculum-only baselines are incomplete or inadequately specified—particularly in the 20B-token setting, where this baseline is omitted. Reported gains over strong alternatives are often marginal, casting doubt on the robustness and scalability of the claimed advantages.

**Reviewer Scores:**

NA

---

### Decision · Program_Chairs · 2026-01-26

Reject